# The Importance of Optimal Hydration in Patients with Heart Failure—Not Always Too Much Fluid

**DOI:** 10.3390/biomedicines11102684

**Published:** 2023-09-30

**Authors:** Andrzej Wittczak, Maciej Ślot, Agata Bielecka-Dabrowa

**Affiliations:** 1Department of Preventive Cardiology and Lipidology, Medical University of Lodz, 90-419 Lodz, Poland; 2Department of Cardiology and Congenital Diseases of Adults, Polish Mother’s Memorial Hospital Research Institute, 93-338 Lodz, Poland; maciej.slot@edu.uni.lodz.pl; 3Faculty of Physics and Applied Informatics, University of Lodz, 90-236 Lodz, Poland

**Keywords:** heart failure, hydration, dehydration, overhydration, congestion, euvolemia, bioimpedance

## Abstract

Heart failure (HF) is a leading cause of morbidity and mortality and a major public health problem. Both overhydration and dehydration are non-physiological states of the body that can adversely affect human health. Congestion and residual congestion are common in patients hospitalized for HF and are associated with poor prognosis and high rates of rehospitalization. However, the clinical problem of dehydration is also prevalent in healthcare and community settings and is associated with increased morbidity and mortality. This article provides a comprehensive review of the issue of congestion and dehydration in HF, including HF guidelines, possible causes of dehydration in HF, confirmed and potential new diagnostic methods. In particular, a full database search on the relationship between dehydration and HF was performed and all available evidence in the literature was reviewed. The novel hypothesis of chronic subclinical hypohydration as a modifiable risk factor for HF is also discussed. It is concluded that maintaining euvolemia is the cornerstone of HF management. Physicians have to find a balance between decongestion therapy and the risk of dehydration.

## 1. Introduction

Heart failure (HF) is a leading cause of morbidity and mortality, and causes high healthcare-related costs, posing a great burden on both patients and society [1]. It is a major public health problem with prevalence estimated to be 1–2% of adults [2]. The prevalence of HF rises with age, from about 1% in those younger than 55 years to more than 10% in those 70 years and older [2]. In wealthy nations, the age-adjusted incidence of HF may be decreasing, likely due to improved cardiovascular disease management, but the overall incidence is increasing as the population ages [2].

Hydration status is defined by the balance between water outputs and water inputs [3]. Physiological homeostasis is disrupted when inputs exceed outputs or vice versa. Both overhydration and dehydration are non-physiological states of the body that can have negative implications for human health. Therefore, clinicians must pay special attention to maintaining optimal hydration in all patients.

The topic of hydration and health is relatively new and still under-researched [4,5]. This statement is all the more true in relation to HF. In the position paper on self-care of HF patients (published by Heart Failure Association of the European Society of Cardiology (ESC)) “optimal fluid and salt management in patients with HF” was listed as a recommendation for future research [6]. In the 2021 ESC Guidelines for the diagnosis and treatment of acute and chronic HF, “evidence on the effects of fluid restriction” was listed as a gap in evidence [2]. A similar statement can be found in the 2022 American Heart Association (AHA) Guideline for the Management of Heart Failure [7]. Thus, there is a need for new studies on hydration in HF.

In this review, we analyze the problem of optimal hydration in patients with HF. Although congestion is inextricably linked to HF, dehydration is also possible in HF patients. In the first part of this article, we briefly summarized the topic of human body water. Next, we reviewed the problem of congestion in HF. In the third part of the article, we described the clinical problem of dehydration. In the sections on congestion and dehydration, we also reviewed diagnostic methods to assess the state of volemia. Finally, in the last part, we comprehensively reviewed the issue of dehydration in HF—including HF guidelines, possible causes of dehydration in HF, and all available evidence in the literature. To our knowledge, we are the first to perform a comprehensive database search on the relationship between dehydration and HF and to review all available evidence. The novel hypothesis of chronic subclinical hypohydration as a modifiable risk factor for HF is also described.

## 2. Human Body Water—Brief Overview

Water is the main constituent of the human body, accounting for approximately 60% of body weight [8]. In an adult, approximately 2/3 of the total body water is found in the intracellular space and the remaining 1/3 is extracellular water. A person weighing 70 kg typically has a total body water volume of approximately 42 L, of which 28 L is inside the cells (intracellular fluid or ICF) and 14 L is outside the cells (extracellular fluid or ECF) [8]. Extracellular fluid is further sub-divided into intravascular and interstitial compartments (5 L and 9 L, respectively) [9].

Euvolemia (from Greek “*eu*-” = “good”/”well”; “volume” and *“-emia*” = “blood”) can be defined as the state of having the normal volume of blood (or more generally, total water) in the body. Both 2021 ESC and 2022 AHA heart failure guidelines use this term without any strict definition [2,7]. It is used, however, in context to describe the “ideal volume status” for the patient and is proposed as an appropriate goal for titration of diuretic therapy in HF patients [10]. Maintaining euvolemia undoubtedly requires optimal hydration. Different conditions lead to an imbalance of water in different compartments of the body [11]. These conditions include fluid overload as well as dehydration. The simplified representation of the relationship between euvolemia, dehydration, and overhydration (congestion) in HF is shown in (Figure 1).

## 3. Congestion/Overhydration in Heart Failure

### 3.1. Congestion in Heart Failure—Overview and Pathogenesis

Fluid overload can be defined as an abnormal buildup of both water and dissolved electrolytes in the body that exceeds physiological levels [12]. Typically, this surplus of fluid collects in the distensible spaces between cells, leading to interstitial edema, and may also result in macroscopic fluid accumulation in the chest (pleural effusions) or abdomen (ascites) [12]. Congestion in HF is defined as fluid accumulation in the intravascular compartment and the interstitial space, resulting from increased cardiac filling pressures caused by maladaptive sodium and water retention by the kidneys [13].

The diagnosis of chronic heart failure (CHF) requires the presence of symptoms and/or signs of HF and objective evidence of cardiac dysfunction [2]. Typical symptoms of HF, such as ankle swelling, breathlessness, orthopnea, and paroxysmal nocturnal dyspnea are inextricably linked with the problem of congestion. Bozkurt et al. proposed a universal definition of HF, which is as follows: “a clinical syndrome with symptoms and/or signs caused by a structural and/or functional cardiac abnormality and corroborated by elevated natriuretic peptide levels and/or objective evidence of pulmonary or systemic congestion” [14].

Reduced cardiac output, increased cardiac filling pressures, impaired natriuresis/diuresis, and excessive peripheral vasoconstriction are the long-established hallmarks of HF disease that result in volume overload. These alterations in regulating volume and cardiac output create a vicious cycle. It begins with diminished cardiac output and the incapacity to sustain a physiological fluid volume, which then contributes to neurohormonal activation. This, in turn, worsens excessive fluid accumulation and dysfunction of the myocardium [15].

Distinction between tissue congestion and intravascular congestion is important as it allows optimized treatment of patients with decompensated HF. In decompensated HF, the initial phase of fluid buildup occurs within the intravascular compartment. Persistently elevated hydrostatic pressures within the capillary vessels result in tissue congestion. While most patients experiencing decompensated HF exhibit a mixture of intravascular and tissue congestion, some experts suggest that one of these two patterns may predominate in certain cases [13].

During an episode of acute decompensated HF, a combination of neurohormonal and hemodynamic factors leads to water and sodium retention by the kidneys [16]. These factors are variously present during different stages and severities of HF and include activation of the renin–angiotensin–aldosterone and natriuretic peptide axes, low cardiac output, the sympatho-sympathetic reflex as a result of cardiac stretch, and probably other unknown contributors [17,18]. Although the activation of these neurohormonal systems maintains cardiovascular homeostasis in the short term, chronic activation of these responses results in hemodynamic stress and exerts deleterious effects on the heart and circulation [19].

#### 3.1.1. Role of Sympathetic Nervous System

Sympathetic nervous system (SNS) activation is one of the key neurohumoral mechanisms that are operative in HF and is robustly associated with arrhythmias, sudden cardiac death, adverse myocardial remodeling, and overall poor prognosis in this population [20]. Physiologically, the SNS plays an important role in determining systemic vascular resistance and in mediating short-term changes in vascular resistance, through changes in circulating norepinephrine as well as activation and deactivation of cardiopulmonary and arterial baroreflexes [21]. Importantly, renal sympathetic nerve activation is a major factor in the anti-natriuretic and vasoconstrictive systems that largely contribute to the avid renal sodium and water retention that characterizes patients with advanced HF [22]. Intravascular congestion that occurs in patients with HF is caused not only by a gradual expansion in plasma volume but also by shifting of fluid from venous reservoirs [13]. The venous system contains approximately 70% of the total blood volume and is approximately 30 times more compliant than the arterial system [23]. Because splanchnic veins contain large numbers of α1 and α2 adrenergic receptors, they are highly sensitive to stimulation by the SNS [23]. Therefore, increase in the SNS activity causes a reduction in venous compliance. This leads to mobilization of fluid from the venous capacitance vessels to the effective circulatory volume [21]. The described dysregulation of blood distribution has been suggested to play an important role in acute intravascular congestion [24].

#### 3.1.2. Role of Renin–Angiotensin–Aldosterone System

The renin–angiotensin–aldosterone system (RAAS) is a central feature in the pathophysiology of HF [25]. It plays a critical role in the homeostasis of extra cellular fluid (ECF) volume, sodium balance, blood pressure, and cardiac performance [22]. Renin is an aspartyl protease produced in the juxtaglomerular cells of the renal afferent arteriole [26,27]. Its secretion is regulated in an opposing manner by changes in blood pressure within the afferent arterioles and by the chloride levels within the tubule fluid, specifically at the macula densa portion of the distal tubule [28]. Renin cleaves 10 amino acids from angiotensinogen to form angiotensin I, which is subsequently cleaved by angiotensin-converting enzyme (ACE) to form angiotensin II [26]. Circulating renin has the potential to directly and/or indirectly modulate HF progression [29]. A pathological increase in plasma renin activity concentration precedes the development of edema in a subset of patients with reduced systolic function with or without symptomatic HF [29]. Angiotensin II, the main effector of the RAAS, controls ECF and urinary sodium excretion [22]. The elevated circulating levels of angiotensin II in HF exacerbate tubular salt and water handling by directly stimulating proximal tubular sodium absorption and indirectly by increasing aldosterone and endothelin release. Furthermore, angiotensin II stimulates thirst despite a typically low serum osmolality [22]. Interestingly, angiotensin II regulates renin secretion, with renin synthesis and secretion being inhibited by increases and stimulated by decreases in angiotensin II concentration [27,30]. Therefore, angiotensin-converting enzyme inhibitors (ACE-I) are thought to stimulate renin secretion by interrupting the feedback inhibition of angiotensin II [30]. Aldosterone, a mineralocorticoid hormone produced by the adrenal gland is the agonist of the mineralocorticoid receptor [31]. It plays a key role in controlling sodium reabsorption in the aldosterone-sensitive distal nephron. Angiotensin II, along with an increase in blood levels of K^+^, stimulates aldosterone secretion. Aldosterone causes the kidney to increase sodium reabsorption back into the bloodstream while simultaneously increasing water reabsorption [32]. This undesirable effect, along with the promotion of vascular remodeling in the heart and other organs, contributes to the progressive nature of HF [33]. To sum up, the RAAS activation causes sodium and fluid retention, which leads to congestion.

#### 3.1.3. Role of Natriuretic Peptide System

The natriuretic peptide system plays a critical role in maintaining cardio–renal homeostasis by counteracting the aforementioned vasoconstrictive, antidiuretic, anti-natriuretic, and tissue remodeling factors/pathways [22]. The system includes two cardiac hormones, atrial natriuretic peptide (ANP) and brain natriuretic peptide (BNP), whose secretion is significantly increased in HF [22]. However, despite their exceptionally elevated circulating levels in plasma, their effects are attenuated [22]. Many mechanisms have been proposed to explain this apparent paradox [34,35]. One of them is the impaired activation of prohormone-ANP (pro-ANP) to ANP caused by the decreased levels of corin, a serine protease in the heart that is primarily responsible for the cleavage-activation of pro-ANP [36,37,38]. Another theory involves alterations in the way natriuretic peptide receptors are expressed and activated, along with the components responsible for their downstream signaling. These changes may be associated with the neurohumoral processes that take place in HF [22,35]. Furthermore, an additional contributing factor may be the increase in the secretion of the less active forms of BNP seen in HF patients, rather than the mature active hormone BNP_1–32_ [22,39]. The commercially available immunoassays lack specificity and measure inactive as well as active forms [35]. In fact, the serum levels of the bioactive BNP_1–32_ might be low in subjects with HF [35,40]. Finally, the peripheral degradation of natriuretic peptides should be emphasized. All natriuretic peptides are degraded through two main processes: (1) internalization and lysosomal degradation mediated by natriuretic peptide clearance receptor (NPR-C) and (2) enzymatic degradation by neutral endopeptidase 24.11 or neprilysin [41]. Neprilysin is particularly important because we have a drug which inhibits its function—sacubitril (in the form of sacubitril/valsartan—known as angiotensin receptor neprilysin inhibitor [ARNI]) [41,42]. Inhibition of neprilysin by ARNI increases the concentration of natriuretic peptides by preventing their degradation, thereby enhancing the beneficial effects of the natriuretic peptide system [22].

#### 3.1.4. Development of Tissue Edema

The development of tissue edema is the result of an imbalance between oncotic and hydrostatic pressures at the level of the interstitium [13]. In physiological conditions, there is a continuous flow of fluid seeping from the capillaries into the interstitial space, and any fluid entering the interstitial area is subsequently drained by the lymphatic system [43]. In case of the right ventricle failure, right ventricle filling is impaired and consequently systemic venous pressure increases [44]. The hydrostatic pressure inside the capillary vessels increases proportionally to the fourth power of the capillary radius, leading to a greater rate of fluid filtration across the capillary walls. Simultaneously, heightened systemic venous pressure might hinder the drainage of lymphatic fluid into the systemic veins. Eventually, a critical point is reached where the lymphatic system’s ability to remove fluid from the interstitial space is overwhelmed, resulting in fluid accumulation [43]. In the event of left ventricular failure, the filling pressure of the left ventricle increases and, as a consequence, so does the pressure in the left atrium and in the pulmonary capillaries. The analogous process of rising hydrostatic pressure in the lung capillaries and impaired lymphatic drainage leads to the accumulation of fluid in the airspaces of the lungs [43]. Initially, the excess filtrate seeks out and fills the peribronchovascular interstitial spaces, which can hold 300–400 mL of fluid (depending on the size of the lung), forming the ‘cuffs’ that are often visible on chest X-rays [45]. When the interstitial spaces are filled with fluid, it begins to enter the alveolar spaces (alveolar edema) [45].

### 3.2. Diagnostics

The 2021 ESC Guidelines for the diagnosis and treatment of acute and chronic HF recommend that patients hospitalized for HF should be carefully evaluated to exclude persistent signs of congestion before discharge [2]. However, there is no established algorithm for the assessment of congestion [46].

#### 3.2.1. Gold Standard

The gold standard for evaluating hemodynamic congestion in HF patients is cardiac catheterization to measure right atrial pressure and pulmonary capillary wedge pressure (PCWP) [46]. Nevertheless, the invasive character of catheterization restricts its regular application in clinical practice. There is no single non-invasive test available that can precisely identify hemodynamic congestion, and the ability to detect congestion poses a diagnostic dilemma since it often occurs before observable clinical symptoms [46].

#### 3.2.2. Symptoms

Dyspnea is one of the most common presenting symptoms of HF. In cases where dyspnea occurs in a supine position, it is referred to as orthopnea. This condition arises due to the movement of fluid from the lower limbs and splanchnic circulation into the cardiopulmonary circulation [47]. This redistribution of fluid increases preload, which promotes an exacerbation of dyspnea. It was proposed that because orthopnea results from increased preload rather than alveolar edema, it should be considered as a symptom of intravascular congestion rather than tissue congestion [13]. Similarly, bendopnea (occurrence of increased dyspnea when bending forwards) seems to be indicative of intravascular congestion [13].

#### 3.2.3. Signs

Jugular venous distention (JVD) is one of the methods used to assess congestion. Distension of the jugular vein provides an indication of right atrial pressure [13]. In current clinical practice, assessment of JVD is usually performed by inspection of the jugular veins and estimating the degree of distension [48]. It was proposed that positive JVD should be interpreted as a sign of intravascular congestion [13]. The precise estimation is difficult, however, and the sensitivity and specificity for positive JVD are poor (57.3% and 43.6%, respectively) [49].

The third heart sound is another sign associated with HF. It is the result of rapid filling of the ventricle in the early part of the diastole and rapid deceleration of blood flow in an already-filled ventricle [50]. It was proposed that it should be considered as a sign of intravascular (or rather intracardiac) congestion [13].

Tissue congestion can be assessed in terms of symptoms and by physical examination. The established indicators are the presence of peripheral edema, ascites, and rales [13]. Pitting edema is highly specific for the presence of interstitial edema [13]. Nevertheless, most clinical symptoms and signs have moderate specificity and poor sensitivity for diagnosing HF as the cause of interstitial edema [51].

#### 3.2.4. Laboratory Tests

Atrial natriuretic peptide (ANP) and B-type natriuretic peptide (BNP) are predominantly produced by the atria and ventricles, respectively, in response to volume/pressure overload [52]. BNP and N-terminal-proBNP have become important diagnostic tools for assessing patients who present acutely with dyspnea [52]. It was proposed that elevated circulating levels of natriuretic peptides are likely to be an indication of intravascular and intracardiac congestion rather than of tissue congestion [13].

#### 3.2.5. Bioelectrical Impedance Analysis

Bioelectrical impedance analysis (BIA) and bioelectrical impedance vector analysis (BIVA) are non-invasive, quick, and affordable methods to accurately assess body composition and fluid status in clinical practice [53,54,55]. Both techniques refer to the measurement of the electrical impedance (Figure 2a) of a biological sample measured across electrodes. In order to study the electrical properties of human tissue, it is necessary to assume some equivalent circuit representing the tissues (Figure 2b) [56]. One of the most commonly used approaches is the so-called Cole model, in which cells forming a tissue are suspended in extracellular fluid (ECF). Each cell consists of intracellular fluid (IFC) and membrane [57]. Both ICF and ECF provide relatively good conductive properties in response to alternative signal excitation. In comparison, cell membrane acts like a capacitor [58,59]. As a result, under the alternating electrical excitations, tissues exhibit complex impedance, which is dependent on tissue structure, anatomy, and frequency of source electrical signal (Figure 2c,d) [60]. Most devices conducting BIA work in the range of 1 kHz–1 MHz with very low amplitude of the current. Due to the tremendous development of electrical engineering and signal processing, this method allows for quick and precise measurement with negligible effect on patient health [61,62].

A number of bioimpedance-related measures including edema index, extracellular body water (ECW), phase angle (PA), resistance, reactance, and hydration index have been used to identify fluid status and degree of fluid congestion [53]. For example, the ECW/TBW (total body water) ratio has been identified in various studies as an indirect measure of overhydration [63,64]. However, there is no consensus on which parameters comprehensively reflect the conditions of HF patients. Moreover, there are many mathematical models implemented in devices, so accuracy of the measurement is dependent not only on quality of hardware, but it is also greatly affected by the method of signal analysis applied by creators of the apparatus [65]. There is some evidence that the combination of BIVA and BNP levels increases the ability to detect fluid overload in HF, improving treatment and preventing further complications (such as worsening of renal function) [66,67,68,69]. According to the 2022 review of the uses of BIA and BIVA in HF patients, BIA can be used to facilitate HF treatments and managements, although BIVA and its derived measures seem to be more accurate values to manage and monitor HF than BIA [53].

#### 3.2.6. Imaging

Recent studies have demonstrated the prognostic role of residual congestion assessed by pre-discharge lung ultrasound (LUS) in patients hospitalized for HF [70,71]. According to Platz et al., LUS may be a useful non-invasive method to monitor dynamic changes in pulmonary congestion in response to treatment, and a presence of residual congestion in patients with chronic HF may identify those at high risk for decompensation [70]. The study by Ceriani et al. showed that residual congestion assessed by LUS at discharge was associated with long-term mortality in patients hospitalized for HF [71].

Chest X-ray (CXR) is one of the most widely used diagnostic imaging techniques [72]. The 2021 ESC HF Guidelines recommend performing a chest X-ray as one of the diagnostic tests in all patients with suspected chronic HF, as it may provide supportive evidence of HF such as pulmonary congestion or cardiomegaly [2]. Indeed, chest radiography can be used to assess the degree of congestion and is mostly specific to pulmonary tissue congestion [13]. Today, due to the commonness of X-rays, the main advantages of this method are relatively low cost and theoretical availability in many medical facilities. However, it is crucial to remember that devices using ionic radiation require special infrastructure and staff experienced in carrying out measurements and analyzing images. Moreover, the systematic review study conducted by Mant et al. showed that CXR is insufficient in diagnosing HF due to the high insensitivity (67–68%) [73]; similar findings were also reported in previous studies [74,75]. According to the 2022 AHA guidelines, chest X-ray should not be used as the only determinant of the specific cause or presence of HF [7].

Chest computed tomography (CT) can be used to diagnose pulmonary edema [76]. This method is considered to be more precise than CXR [76]. Indeed, increased density on high-resolution pulmonary CT scans correlates well with lung weight and has been suggested as a gold standard for the assessment of pulmonary interstitial edema [13]. On the other hand, the certain drawbacks are relatively high exposure to ionic radiation and cost of apparatus making it unreachable for many medical centers [77].

### 3.3. Treatment Overview

#### 3.3.1. Diuretics

According to the 2021 ESC Guidelines for the diagnosis and treatment of acute and chronic HF, loop diuretics are recommended to reduce the signs and/or symptoms of congestion in patients with HFrEF, HFmrEF, and HFpEF (heart failure with reduced, mildly reduced, and preserved ejection fraction, respectively) [2]. The goal of diuretic treatment is to attain and sustain euvolemia using the smallest possible diuretic dosage [2]. The guidelines emphasize that patients should be educated to independently regulate their diuretic dosage by monitoring symptoms or signs of congestion and by measuring their weight daily [2].

#### 3.3.2. Other Drugs

Of course, all patients with HF should be treated with medications that have been shown to improve survival and reduce the risk of HF hospitalizations, namely angiotensin-converting enzyme inhibitors (ACE-I), beta-blockers, mineralocorticoid receptor antagonists (MRA), and sodium-glucose co-transporter 2 (SGLT2) inhibitors (recommendations depend on HF phenotype) [2]. By counteracting multiple pathophysiological processes responsible for the clinical picture of HF, these drugs also reduce congestion.

#### 3.3.3. Sodium-Glucose Co-Transporter 2 Inhibitors (SGLT2 Inhibitors)

SGLT2 inhibitors have emerged as a new foundational therapy in patients with HF, initially only in HF with reduced ejection fraction (HFrEF) [78]. Currently, SGLT2 inhibitors (dapagliflozin or empagliflozin) are also recommended in patients with mildly reduced EF (HFmrEF) and with preserved EF (HFpEF) to reduce the risk of HF hospitalization or cardiovascular death (class I, level A) [79]. Large-scale randomized controlled trials (RCTs) of SGLT2 inhibitors have shown significant cardiovascular and renal benefit across various subgroups [78]. These drugs significantly reduced the risk of all-cause mortality, cardiovascular mortality, HF hospitalization, and renal outcomes [78]. SGLT2 inhibitors cause natriuresis by inhibition of glucose transport, which is driven by concurrent sodium transport in the proximal convoluted tubule in the kidney [13]. The diuretic/natriuretic properties of SGLT2 inhibitors may offer benefits in reducing congestion and may allow a reduction in loop diuretic requirement [2,80]. HFrEF pre-clinical studies have shown that SGLT2 inhibitors treatment may attenuate edema formation not only through the stimulation of natriuretic and osmotic/diuretic effects, but also through improvement in overall cardiac function, and the suppression of maladaptive cardiac remodeling, chronic inflammation, oxidative stress, and endothelial dysfunction [81]. Clinical HFrEF studies also suggest that SGLT-2 inhibitors treatment may attenuate the pathological salt-water retention [82]. Interestingly, one study hypothesized a larger role for aquaresis, suggesting the possibility of using SGLT2 inhibitors in the treatment of tissue congestion, similar to the use of vasopressin antagonists [13,83]. According to the recent review by Biegus et al., there are several potential mechanisms that can individually or collectively facilitate decongestion mediated by SGLT2 inhibitors [84]. These include: (1) osmotic diuresis and natriuretic effect; (2) restoration of glomerular filtration and correction of hyperfiltration; (3) interstitial drainage; (4) intracardiac pressure reduction; (5) intravascular volume contraction; and (6) hemoconcentration [84].

#### 3.3.4. Angiotensin-Converting Enzyme Inhibitors (ACE-Is)

ACE-Is were the first class of drugs shown to reduce mortality and morbidity in patients with HFrEF and are recommended in all patients unless contraindicated or not tolerated [2]. RAAS blockers produce beneficial outcomes in HFrEF by affecting both cardiac and vascular functions. With respect to the kidneys, RAAS blockers stimulate diuretic and natriuretic responses, particularly in the initial phases of their use [22].

#### 3.3.5. Mineralocorticoid Receptor Antagonists (MRAs)

Mineralocorticoid receptor antagonists (MRAs) are recommended, in addition to an angiotensin-converting enzyme inhibitors (ACE-I) and a beta-blocker, in all patients with HFrEF to reduce mortality and the risk of HF hospitalization [2]. MRAs inhibit the effects of aldosterone, thereby increasing sodium excretion and potassium retention [13]. This effect may be beneficial in reducing congestion. The addition of natriuretic doses of MRAs may offer an alternative to the intensification of loop diuretic regimen in treating congestion in HF patients [85]. In patients with left ventricular dysfunction after myocardial infarction, patients receiving 25–50 mg of eplerenone daily had significantly greater reductions in body weight and higher hemoconcentrations than patients receiving placebo, suggesting a diuretic effect of eplerenone [86]. Doses of ≥100 mg of spironolactone daily have been shown to increase natriuresis [87,88].

#### 3.3.6. Sacubitril/Valsartan (ARNI)

Sacubitril/valsartan (ARNI), a medication that combines an angiotensin receptor blocker and a neprilysin inhibitor, demonstrated superior efficacy compared to ACE inhibitors in lowering the risks of death and hospitalization due to HF [89]. The inhibition of neprilysin through ARNI leads to elevated levels of natriuretic peptides by preventing their breakdown. Consequently, this triggers natriuresis, lowers blood pressure, and inhibits the cardiac myocyte hypertrophy, fibrosis, and apoptosis [22]. The utilization of ARNI may potentially result in a decreased need for loop diuretics [2,90].

#### 3.3.7. Beta Blockers

In patients with HFrEF, when given with ACE-I and diuretic, beta-blockers have been shown to reduce mortality and morbidity [2]. Since activation of the β1 receptor stimulates renin secretion, it is possible that the beneficial effects of β-blockers at both the cardiac and renal levels are partially attributed to attenuation of renin secretion [22].

### 3.4. Congestion-Self-Care and Ambulatory Care of Heart Failure Patients

Symptom monitoring, along with medication adherence and preventive behaviors (such as sodium restriction), is the foundation of HF self-care maintenance [91]. The burden of daily assessment for signs and symptoms of worsening HF falls largely on the patient and the ambulatory care provider [91]. Early detection of a deterioration in HF symptoms is important in order to prevent hospital admission and increased mortality [6]. The authors of the practical management recommendations from the Heart Failure Association of the ESC recommend special guidance for self-care monitoring and self-care management of patients with HF [6]. Patients are encouraged to weigh themselves daily, take their medications as prescribed, and keep their appointments. If patients notice worsening symptoms of HF (such as dyspnea, leg swelling, weight gain of ≥2 kg in 2–3 days), they are advised to contact an HF nurse or general practitioner [6]. The ESC HF guidelines state that in the case of increasing dyspnea or edema, or sudden unexpected weight gain of >2 kg in 3 days, patients may also increase their diuretic dose [2]. Nevertheless, more evidence on the optimal models for follow-up of stable HF patients is needed [2].

### 3.5. Congestion/Overhydration in Heart Failure—Summary

In conclusion, residual congestion is frequently found in patients who are hospitalized for HF and is associated with a poor prognosis and a high rate of (short-term) rehospitalization [13]. HF guidelines recommend that patients hospitalized for HF should be carefully evaluated to exclude persistent signs of congestion before discharge and to optimize oral treatment [2].

## 4. Dehydration—Overview

“Dehydration” is a term which, in clinical use, refers to a deficiency in total body water [92]. Although no standard means of defining its presence or severity exists, it appears to be both prevalent and costly within the healthcare setting. In the 2015 prospective cohort study performed on patients aged ≥65 years who were admitted acutely to a large UK teaching hospital, 37% of participants were dehydrated [93]. Of 370,758 patients in the 2004 United States National Hospital Discharge Survey, there were 518,000 hospitalizations primarily due to dehydration, incurring healthcare costs in excess of 5 billion dollars [94].

There is no universally accepted definition for dehydration in humans [92]. The lack of international consensus on the definition and diagnosis of dehydration complicates an assessment of its prevalence and clinical research in this field. Traditionally, the medical literature has categorized total body water fluid loss into two primary forms: (1) dehydration, which involves the reduction of body water primarily from within the intracellular compartments, and (2) volume depletion, which pertains to a decrease in extracellular fluid [95]. While this distinction makes sense from the physiological point of view, most clinicians tend to use the term dehydration for any loss of total body water. Therefore, Thomas et al. suggested that dehydration should be defined as a complex condition resulting in a reduction in total body water [95]. This can be due primarily to a water deficit (water loss dehydration) or both a salt and water deficit (salt loss dehydration) [95]. The 2018 ESPEN (European Society for Clinical Nutrition and Metabolism) guideline on clinical nutrition and hydration in geriatrics differentiate low-intake dehydration and volume depletion [96]. Low-intake dehydration occurs when there is an insufficient intake of pure water, leading to the loss of both intracellular and extracellular fluids. This results in an increase in osmolality in both compartments, both intracellular and extracellular. Volume depletion, on the other hand, is caused by excessive loss of fluid and salts, especially sodium. The primary loss is in the extracellular fluid, and serum osmolality will typically remain normal or decrease [96]. It is important to distinguish between “dehydration” and “hypovolemia”. The term “hypovolemia” should be exclusively used to describe the reduction of intravascular volume, which can occur as a result of dehydration (most frequently in cases of isotonic dehydration) but should not be used interchangeably with it [92].

### 4.1. Objective Methods for Diagnosing Dehydration

According to the 2019 multidisciplinary consensus on dehydration, the gold standard for determining dehydration is a direct measurement of serum/plasma osmolality [92]. In the absence of excessive electrolyte fluctuations, plasma osmolality (pOsm) can function as a reliable indicator of aberrant fluid status, encompassing both states of dehydration and fluid overload. Since pOsm can be employed to determine fluid deficit based on a single measurement in an individual (using reference intervals from the general population), it becomes a valuable tool for assessing hydration status [92]. In the study by Cheuvront et al., pOsm demonstrated a 90% sensitivity and 100% specificity for detecting dehydration associated with a 2% fall in body mass [97]. However, it is important to note that isotonic dehydration cannot be reliably detected by changes in plasma osmolality [92]. According to the 2019 multidisciplinary consensus on dehydration, measured plasma osmolality > 300 mOsm/kg classifies a person as hyperosmolar and plasma osmolality ≤ 280 mOsm/kg classifies a person as hypo-osmolar [92]. As direct laboratory measurement of plasma osmolality requires advanced equipment and competent technicians, the consensus on dehydration recommend the use of the Khajuria–Krahn equation as a surrogate [92]. The calculated plasma osmolarity (pOsm_c_) can be determined by using the formula from Figure 3 [92,98]. It is recommended to use this equation as a screening test for hypertonic dehydration. ESPEN guidelines on clinical nutrition and hydration in geriatrics recommend the use of the same equation (with an action threshold of >295 mmol/L) to screen for low-intake dehydration in older persons [96]. An elevated calculated osmolarity can be verified by direct measurement of osmolality [92].

It is possible to assess hydration status through total body water (TBW) measurement; however, it is difficult outside well controlled experimental settings. The gold standard for TBW measurement is tracer dilution techniques, which do not have utility in everyday clinical practice [92]. According to the 2015 Cochrane systematic review, a bioelectrical impedance analysis of total body water was not found to be useful as an individual way of assessing presence or absence of dehydration in older people [99]. However, another bioelectrical impedance analysis parameter—resistance at 50 kHz—was found to be a test which shows some potential ability to diagnose water-loss dehydration as a standalone test [99].

### 4.2. Clinical Diagnosis of Dehydration—Signs and Symptoms

Dehydration cannot be clinically defined by a single symptom, sign, or laboratory value [95]. Serial measurements of weight, laboratory values (creatinine, serum osmolarity, urinary sodium, BUN (blood urea nitrogen)), and physical signs (tongue dryness and furrows, dry mucous membranes, low urine output, speech difficulty, confusion, sunken eyes, low blood pressure, and increased pulse rate) are reported to be needed to make an accurate diagnosis. Classical physical signs are too insensitive and nonspecific to draw conclusions about dehydration [100]. Similarly, according to the 2019 multidisciplinary consensus on dehydration, clinical signs and symptoms of dehydration in adults are subtle and may be unreliable. They should not be used in isolation for detecting abnormalities in hydration (to approximate volume or osmolality) [92]. Nevertheless, the consensus underlines that the volume status of all patients should be assessed regardless of their plasma osmolality [92]. The experts advise utilizing the National Institute for Health and Care Excellence (NICE) guidelines on intravenous fluid therapy in adults in hospital during the initial evaluation and resuscitation of patients in an acute care environment [92,101]. According to the NICE guidelines [101], indicators that a patient may need urgent fluid resuscitation are as follows: systolic blood pressure less than 100 mmHg; heart rate more than 90 beats per minute; capillary refill time longer than 2 s or cold to touch peripheries; respiratory rate more than 20 breaths per minute; and National Early Warning Score (NEWS) of 5 or more (NEWS is a scoring system developed by The Royal College of Physicians to improve the detection of and response to clinical deterioration in patients with acute illness [102]).

The 2015 Cochrane systematic review on clinical symptoms, signs, and tests for identification of water-loss dehydration in older people found that none of the potential signs/markers of hypertonic dehydration had adequate sensitivity and specificity [99]. A total of 24 of the included studies assessed 67 tests (including mouth dryness, skin turgor, capillary refill, thirst, body temperature, confusion, urine concentration, and bioelectrical impedance) in adults aged at least 65 years old. Among all included tests, only expressing fatigue, observed reduced oral fluid intake, and one bioelectrical impedance analysis parameter (resistance at 50 kHz) correlated with dehydration to some degree in some studies, but often with low diagnostic accuracy. No clinical signs were consistently diagnostically accurate in more than one study. The authors concluded that there is no clear evidence for the use of any single clinical symptom, sign, or test of water-loss dehydration in older people [99].

Main clinical findings of dehydration and overhydration (congestion) are summarized in Table 1.

### 4.3. Treatment Overview

Despite all the aforementioned problems with evidence-based dehydration diagnosis, the condition is diagnosed and treated on a daily basis by physicians all over the world. Since “prevention is better than cure”, it should be emphasized that adequate fluid intake is an essential element of the human diet. Daily water intake is essential to counterbalance the daily losses incurred through processes such as respiration, transpiration, urination, and defecation [96]. According to the European Food Safety Authority (EFSA) an adequate total water intake for females is 2.0 L/day and for males 2.5 L/day [103]. “Total water intake” includes all consumed water from a combination of drinking water, beverages, and food. EFSA recommended that the same values are adequate intakes for the elderly [103]. The authors of the ESPEN guideline on clinical nutrition and hydration in geriatrics assumed that 80% of the total water intake comes from consumed fluids and therefore recommended that: “Older women should be offered at least 1.6 L of drinks each day, while older men should be offered at least 2.0 L of drinks each day unless there is a clinical condition that requires different approach” [96]. Heart failure is generally considered to be one of these conditions—according to the 2021 ESC guidelines, “a fluid restriction of 1.5–2 L/day may be considered in patients with severe HF/hyponatremia to relieve symptoms and congestion” [2] (however, this intervention is controversial—see below).

Of course, given the high prevalence of dehydration in hospitalized patients, appropriate treatment is essential. Treatment for volume depletion is aimed at replacing lost water and electrolytes and involves the administration of isotonic fluids. ESPEN recommends that older adults with mild/moderate/severe volume depletion should receive isotonic fluids orally, nasogastrically, subcutaneously, or intravenously [96]. Intravenous (IV) fluid administration is one of the most commonly used therapies in hospitals, and when used appropriately, it improves outcomes [104]. In 2013, NICE published evidence-based guidelines on intravenous fluid therapy in adults in hospital [101]. The document describes in detail the general principles for managing intravenous fluid therapy. For example, the authors recommend that if a patient needs IV fluid resuscitation, physicians should utilize crystalloid solutions that encompass sodium concentrations falling within the range of 130–154 mmol/L, administering a 500 mL bolus over a duration of less than 15 min [101]. It is important to emphasize that patients receiving IV fluid therapy have to be adequately monitored in order to avoid volume overload, especially HF patients.

### 4.4. Dehydration—Summary

In conclusion, it is important to emphasize that dehydration is prevalent within the healthcare setting and as well as in the community and appears to be associated with increased morbidity and mortality [92]. As such, dehydration is a major challenge for clinicians and a significant public health risk. It has a complex, variable pathophysiology that can lead to non-specific clinical presentations that make assessment difficult [92]. Therefore, clinicians must pay close attention to the hydration status of all patients and identify situations that require appropriate management.

## 5. Dehydration and Heart Failure

Although congestion is inextricably linked to HF, dehydration is also possible in HF patients. According to some authors, it is even a common occurrence [105]. In the 2021 ESC HF guidelines, dehydration is mentioned in the section on the use of diuretics [2].

### 5.1. Dehydration in Heart Failure—Possible Causes

#### 5.1.1. Diuretics

Given the pivotal role of congestion in HF, diuretics are a cornerstone of therapy in HF [106]. Adverse effects for loop diuretics typically occur from electrolyte imbalances secondary to the diuresis effects, which include, among others: hyponatremia, hypokalemia, hypochloremia, hypomagnesemia, and dehydration [107]. For this reason, the 2021 ESC HF guidelines emphasize that the aim of diuretic therapy is to achieve and maintain euvolemia with the lowest diuretic dose [2]. The diuretic’s dose may need to be increased or decreased according to the patient’s volume status as excessive diuresis is more dangerous than edema itself. Physicians must be aware of the possibility of diuretic-induced hypovolemia and act accordingly: “Assess volume status; consider a diuretic dosage reduction” [2]. Moreover, SGLT2 inhibitors may also intensify the diuresis, particularly when accompanied by sacubitril/valsartan and diuretic therapy [2]. The authors of the guidelines emphasize that diuretic doses along with fluid intake should be balanced in order to avoid dehydration, symptomatic hypotension, and prerenal renal failure [2]. Interestingly, the quality of the evidence regarding diuretics is poor and their effects on morbidity and mortality have not been studied in randomized controlled trials [2]. The 2020 meta-analysis of observational studies found that loop diuretics, especially in high doses, were associated with increased all-cause mortality and HF hospitalization rates [88]. Such data suggest that the lowest possible diuretic doses should always be used to maintain euvolemia in patients.

#### 5.1.2. Fluid Loss

Dehydration in HF is also possible in conditions that promote increased fluid loss. In the “Patient education and self-care” table of 2021 ESC HF guidelines, the authors recommend increasing fluid intake during periods of high heat/humidity and/or nausea/vomiting [2]. In such situations, not only fluid intake but also diuretic dosage should be adjusted accordingly. Indeed, in the review article by the Balmain et al., the authors emphasize that monitoring of fluid status should be reviewed regularly, as the need for fluid restriction and diuretic use may change from season to season with changes in temperature and patient activity [108]. Furthermore, vomiting and prolonged diarrhea are known causes of isotonic dehydration [92].

#### 5.1.3. Fluid Intake Restriction

Advising patients with chronic HF to restrict fluid intake has been common clinical practice for many decades [109]. The intention is to prevent congestion; however, currently, this intervention is viewed as controversial. Scant and low-quality evidence indicates that restricting fluid intake could potentially lower the risk of HF hospitalization. However, contrasting reports suggest that such restrictions do not yield favorable outcomes and may negatively impact patients’ quality of life [6,110,111]. Nevertheless, in the “Patient education and self-care” table of 2021 ESC HF guidelines, the following recommendation is included: “A fluid restriction of 1.5–2 L/day may be considered in patients with severe HF/hyponatremia to relieve symptoms and congestion” (although it is a general recommendation, no level of evidence is provided) [2]. All in all, significant restriction of fluid intake can undoubtedly be a cause of dehydration, even in HF patients, especially under conditions of high heat (as mentioned above).

#### 5.1.4. Summary

Possible causes of dehydration in HF are summarized in Figure 4.

### 5.2. Heart Failure and Dehydration in the Literature

#### 5.2.1. Methodology

To comprehensively review the relationship between dehydration and HF, we decided to perform a full database search on this topic. Using the Preferred Reporting Items for Systematic reviews and Meta-Analyses (PRISMA) statement methodology [112], we performed the search in the MEDLINE database (PubMed) and Web of Science. The following search query was used in both databases: “heart failure AND dehydration”. No filters (such as year of publication or article type) were used during the search. The search was performed on 15 June 2023. After removing duplicate records, we screened the abstracts of all remaining articles. All publications with promising content on the relationship between dehydration and HF were sought for retrieval. During the evaluation of the full articles, we excluded all documents that did not contain data on dehydration in HF. Articles in languages other than English were also excluded. Additional eligible articles were added by citation search. The PRISMA flow diagram is shown in (Figure 5).

Below, we review all included articles. Out of 20 publications included, 4 articles are case reports [113,114,115,116], 13 are described in the Section 5.2.3 [105,117,118,119,120,121,122,123,124,125,126,127,128], and 3 in the Section 5.3 [129,130,131] (titled “Heart failure and hypohydration”).

#### 5.2.2. Case Reports

During the search, we identified four case reports on dehydration in HF [113,114,115,116]. A summary of these articles is provided in Table 2.

#### 5.2.3. Review of Included Articles

In 2020, the French Society of Cardiology published the expert consensus on the practical management of worsening renal function in outpatients with HF and reduced ejection fraction [105]. The authors emphasize that congestion, hypotension, and dehydration are the main drivers of worsening renal function in patients with HF. According to the experts, dehydration frequently occurs in individuals with HF and is primarily caused by aggressive diuretic therapy. It is particularly prevalent in older patients with HF, notably those with HFpEF. Additionally, factors outside of the clinical setting, such as inadequate fluid intake, exposure to high temperatures leading to increased perspiration, fever, and gastrointestinal issues, can contribute to dehydration. Dehydration, in turn, results in reduced blood flow to the kidneys, leading to prerenal renal failure [105]. The issue of declining kidney function due to excessive dehydration resulting from overly aggressive diuretic therapy (along with passive renal congestion or increased venous pressure that demands higher diuretic doses) is also described by Galas et al. [117]. In the article by Vecchis et al., the authors emphasize that the use of intensive intravenous diuretic therapy carries the risk of causing tubular or glomerular injury and exacerbating often pre-existing renal dysfunction, especially if excessive doses of loop diuretics are inadvertently administered. Such treatment may result in hypotension, hypoperfusion, and/or relative dehydration in patients with decompensated congestive HF [118].

Watanabe et al. emphasized that maintenance of euvolemia with diuretics at appropriate dosage is critical in HF patients with chronic kidney disease (CKD); however, it is difficult to determine the optimal cessation point of decongestion because no reliable marker of volume status to identify euvolemia exists [119]. The authors conducted a study aiming to investigate whether fractional excretion of urea nitrogen (FEUN) is a surrogate marker of volume status for risk stratification in HF patients with CKD. The study showed that in HF patients with CKD, FEUN can serve as a potential indicator of volume status when assessing the risk of readmission for HF after discharge. A low FEUN value (FEUN ≤ 32.1) may indicate intravascular dehydration, while a high FEUN value (FEUN > 38.0) may suggest lingering congestion. Both of these conditions were identified as independent risk factors for HF readmission. The authors concluded that FEUN may be useful to determine euvolemia and guide decongestive treatment in HF patients with CKD [119].

Akhtar et al. published an article with recommendations for patients with cardiovascular disease undergoing Ramadan fasting [120]. The authors warn of the risk of fasting-associated dehydration. With regard to HF, an expert panel on the safety of SGLT2 inhibitors in patients observing Ramadan fasting is mentioned. The experts recommend that SGLT2 inhibitors should be initiated at low dose in the evening at least 1 month before Ramadan and that fluid intake should be increased during non-fasting hours to minimize the risk of dehydration [132].

Chuda et al. conducted a study in 113 patients with HF, of whom 23 (20%) had atrial fibrillation (AF) and 90 (80%) were in sinus rhythm [121]. Patients with AF had significantly lower TBW% (percentage of total body water) measured by bioelectrical impedance analysis (45.7 vs. 50.0%; *p* = 0.022). In a multiple logistic regression model, lower TBW% in body mass analysis (OR (odds ratio) 0.90 per unit increase, *p* = 0.03) was independently associated with AF in patients with HF. The authors concluded that the potential association of lower TBW% and higher body mass index (BMI) with AF may indicate the need for optimal hydration levels and body composition for patients with HF [121].

Several authors noted that elevation of blood urea nitrogen (BUN), potentially indicative of dehydration, is predictive of mortality in decompensated and chronic HF [92,122,123]. In a research project encompassing 263 hospitals throughout the United States and involving a patient cohort of over 65,000 individuals, it was discovered that the most effective predictor of mortality among patients admitted with decompensated HF was the admission level of BUN (which was superior to low systolic blood pressure). BUN level of 43 mg/dL or higher (≥15.35 mmol/L) at admission was the best single discriminator between hospital survivors and non-survivors [122].

In 2016, Núñez et al. published a study on bioelectrical impedance vector analysis (BIVA) and clinical outcomes in patients with acute HF [124]. The authors included 369 consecutive patients discharged from the cardiology department with a diagnosis of acute HF and grouped them into three categories (hyper-hydration, normo-hydration, and dehydration) on the basis of BIVA hydration status. The study showed that 16.8% of patients were dehydrated, 45% were normo-hydrated, and 38.2% were hyper-hydrated. Normo-hydrated patients exhibited the lowest mortality rate (15.1 to 1.43% per 10 person-years); it was intermediate for those with dehydration (19.4 to 2.24% per 10 person-years) and highest for patients with hyper-hydration (30.5 to 3.28% per 10 person-years). The authors concluded that their results were consistent with evidence that fluid overload identifies patients with more advanced disease and a higher risk of death and hospitalization. However, they emphasized that pre-discharge dehydration may identify a subgroup of patients in whom fluid overload is not the main pathophysiologic mechanism causing decompensation, and therefore this subgroup of patients may be at higher risk of adverse effects associated with excessive/inappropriate decongestive treatments [124].

In the study by Ikuta et al. conducted in 746 patients with acute HF, low body mass index (BMI) was independently associated with higher mortality and hospitalization due to dehydration [125]. Asada et al. evaluated the association between intrathoracic impedance and volume loss events in 36 patients with chronic HF and implanted Cardiac Resynchronization Therapy-Defibrillator (CRT-D) or implantable cardioverter-defibrillator (ICD). The study showed that a large positive deviation of thoracic impedance reflected dehydration and bleeding events with a high positive predictive value in severe HF patients [126].

Waldréus et al. conducted a study to compare the intensity of thirst in patients with and without HF and to assess the relationship of this symptom with health-related quality of life and indices of fluid balance [127]. The HF patients reported significantly more intensive thirst compared with those in the control group (perceived thirst was assessed using the visual analog scale). There was no statistically significant relationship between thirst and health-related quality of life, which was low overall. Interestingly, the urine color chart indicated the presence of dehydration (urine color ≥ 3 during the standardized visual assessment) in approximately 60% of patients in both groups. It is worth mentioning that urine color has been validated as a simple and practical index of dehydration in sportsmen [133].

In the article by Robinson et al., the authors address the problem of misdiagnosis of dehydration in the clinical setting [128]. They note that the clinical diagnosis of dehydration is often made when the ordered laboratory tests show an elevated blood urea nitrogen (BUN) and the BUN to creatinine ratio (BUN/Cr). However, this ratio is not specific to dehydration and is abnormal when disease affects the vascular volume as perceived by the kidney, regardless of the status of total body water. One such situation described in the article is congestive HF. Despite the expansion of total body fluid volume, congestive HF activates the same homeostatic mechanisms as dehydration. The primitive mechanism of redistribution of reduced cardiac output to the heart and brain at the expense of all other organs (including the kidney) reduces renal perfusion via sympathetic and angiotensin mediators. This reduces afferent arteriolar flow and effective glomerular perfusion. This low flow state then allows preferential reabsorption of urea with water and salt, resulting in an increase in the BUN/Cr ratio [128].

### 5.3. Heart Failure and Hypohydration

It had been hypothesized that chronic lifelong subclinical hypohydration, which manifests as an elevated serum sodium concentration within the normal range, contributes to the onset of HF [129]. This hypothesis originated from studies utilizing a mouse model of mild lifelong water restriction that elevated serum sodium concentration by 5 mmol/L [130,134,135]. In the study by Allen et al., the authors initiated a mild, lifelong water restriction in mice when they were one month old. This restriction was accomplished by feeding them a gel-based diet consisting of 30% water and 70% dry food, without any supplementary water provided [130]. This type of water restriction protocol was previously proved to increase serum sodium by 5 mmol/L and increase expression of NFAT5 (nuclear factor of activated T-cells 5), a master regulator of hypertonic response [135]. The mice adapted to the induced water restriction and showed no visible distress. Despite the adaptation, the water-restricted (WR) mice had elevated urine osmolality and slightly elevated hematocrit levels, which confirmed a chronic state of mild dehydration. No differences in growth rate and weight were noted during the first year of life; however, after 1 year, the weight of the WR mice started to decrease. The lifespan of WR mice was also shortened by about 6 months compared with control mice that had free access to water. A detailed analysis of body composition and food intake showed that water restriction caused stable metabolism adaptation characterized by increased energy expenditure. Even after 5 months of continuous water restriction, the chronically WR mice still exhibited slightly elevated blood levels of vWF (von Willebrand factor) and D-Dimer. This suggests that a low-grade prothrombotic condition persisted throughout the entire duration of the water restrictions. To test the level of inflammation, the authors measured the blood concentration of IL-6 (interleukin 6); after 5 months of water restriction, the blood level of IL-6 was low with no difference between the control and WR mice; however, the level of IL-6 increased in both groups by age 14 months and to a significantly larger degree in WR mice. This indicated that age-related proinflammatory changes were accelerated by chronic hypohydration. Examination of postmortem tissues revealed a higher degree of renal glomerular injury and cardiac fibrosis in the WR mice [130].

Lang et al. wrote an article on the pathophysiology of suboptimal fluid intake, in which the authors stated that elevated copeptin levels (reflecting elevated vasopressin levels) are associated with increased risk of metabolic syndrome, type 2 diabetes, hypertension, coronary artery disease, and HF [131]. The association between high copeptin levels and HF came from the study by Enhörning et al. [136].

On the basis of results from animal studies, Dmitrieva et al. decided to test the hypothesis of an association between chronic hypohydration and HF by analyzing data from Atherosclerosis Risk in Communities (ARIC) study [129]. ARIC is an ongoing population-based prospective cohort study in which 15,792 45- to 66-year-old black (African American) and white men and women were enrolled from four US communities in 1987–1989 and followed up for more than 25 years [137]. The authors used serum sodium measured at Visits 1 and 2 that took place 3 years apart as a measure of study participants hydration habits. Participants without water balance dysregulation were selected: serum sodium within normal range (135–146 mmol/L), not obese, not diabetic, and free of HF at baseline (N = 11,814). The study revealed that in time-to-event analysis, there was a 39% increase in the risk of HF if serum sodium levels exceeded 143 mmol/L in middle-aged individuals, corresponding to a 1% deficit in body weight water (hazard ratio 1.39, 95% confidence interval (CI) 1.14–1.70). In a retrospective case-control analysis conducted on individuals aged 70 to 90 years who attended Visit 5 (N = 4961), having serum sodium levels ranging from 142.5 to 143 mmol/L was associated with a 62% increase in the likelihood of being diagnosed with left ventricular hypertrophy (LVH) (odds ratio (OR) 1.62, 95% CI 1.03–2.55). Furthermore, having serum sodium levels above 143 mmol/L was linked to a 107% increase in the odds of LVH diagnosis and a 54% increase in the odds of HF (OR 2.07, 95% CI 1.30–3.28 and OR 1.54, 95% CI 1.06–2.23, respectively). The authors concluded that chronic subclinical hypohydration could be considered as a new modifiable risk factor for HF and LVH that was not considered before. They suggest adding recommendations for optimal fluid intake in the same portfolio with recommendations for salt intake as good hydration could be a preventive measure for long-term cardiovascular health outcomes [129].

### 5.4. Dehydration-Self-Care and Ambulatory Care of Heart Failure Patients

As shown in the literature review above, there is a paucity of evidence regarding appropriate ambulatory care and self-care of HF patients with respect to dehydration. In the table titled “Changes for patients with heart failure to self-monitor” in the position paper on self-care of HF patients (ESC 2021), the following symptoms/signs are linked to dehydration: thirst, dizziness, and weight loss [6].

As described in Section 5.1, there are three main causes of dehydration in HF patients: inappropriate dosage of diuretics, fluid loss, and fluid intake restriction. The guidelines emphasize that the aim of diuretic therapy is to achieve and maintain euvolemia with the lowest diuretic dose [2]. Physicians must be aware of the possibility of diuretic-induced hypovolemia and always consider the risk of dehydration. As for excessive fluid loss, the guidelines recommend to increase fluid intake during periods of high heat/humidity and/or nausea/vomiting [2]. Advising patients with chronic HF to restrict fluid intake as a self-care maintenance technique has been a common clinical practice for many decades; however, currently, this intervention is viewed as controversial (see above).

There is no doubt that there is still a clear need for more evidence on this topic, for example on the effects of fluid restriction [2].

## 6. Conclusions

In this review, we have summarized the current knowledge about optimal hydration in HF. Of course, congestion is inextricably linked to HF, but we must not forget that dehydration is possible in a patient diagnosed with this condition. Therefore, physicians have to find a balance between decongestion therapy and the risk of dehydration. It is important to emphasize that the aim of diuretic therapy is to achieve and maintain euvolemia with the lowest diuretic dose. Patients should be trained to self-adjust their diuretic dose based on monitoring of symptoms/signs of congestion (e.g., dyspnea, edema) and daily weight measurements (increase the dose in case of sudden unexpected weight gain of >2 kg in 3 days) [2,6]. On the other hand, patients should be aware that diuretic dose may need to be decreased (and fluid intake increased) if there is fluid loss (e.g., due to diarrhea/vomiting, excessive sweating) [2]. The problem of balancing fluid inputs and outputs is presented in (Figure 6).

The topic of hydration and HF is under-researched. Our database search on dehydration in HF revealed a lack of good quality evidence on this topic. There is a notable lack of hydration-oriented randomized controlled trials. As more detailed evidence-based guideline recommendations are undoubtedly needed, further research on hydration in HF is awaited. For example, results from the FRESH-UP Study (Fluid REStriction in Heart Failure vs. Liberal Fluid Uptake) may answer the question of whether it is beneficial to recommend fluid restriction in patients with chronic HF [109]. Another issue is the availability of reliable prognostic models in HF patients, which allow physicians to develop realistic prognostic expectations and select appropriate therapy and monitoring methods [138,139]. Hydration status should be included in such models.

It is also important to emphasize that available methods to assess hydration status in HF patients are imperfect and not validated. There is a clear need for an evidence-based, affordable, quick, and reliable method to assess hydration status in this population. Such a method would enable physicians to personalize the management of chronic HF. Optimal hydration undoubtedly reduces the risk of HF worsening and HF rehospitalization.

In conclusion, maintaining euvolemia is the cornerstone of heart failure management. Optimal hydration and appropriate pharmacotherapy are crucial elements of heart failure patient care.

## Figures and Tables

**Figure 1 biomedicines-11-02684-f001:**
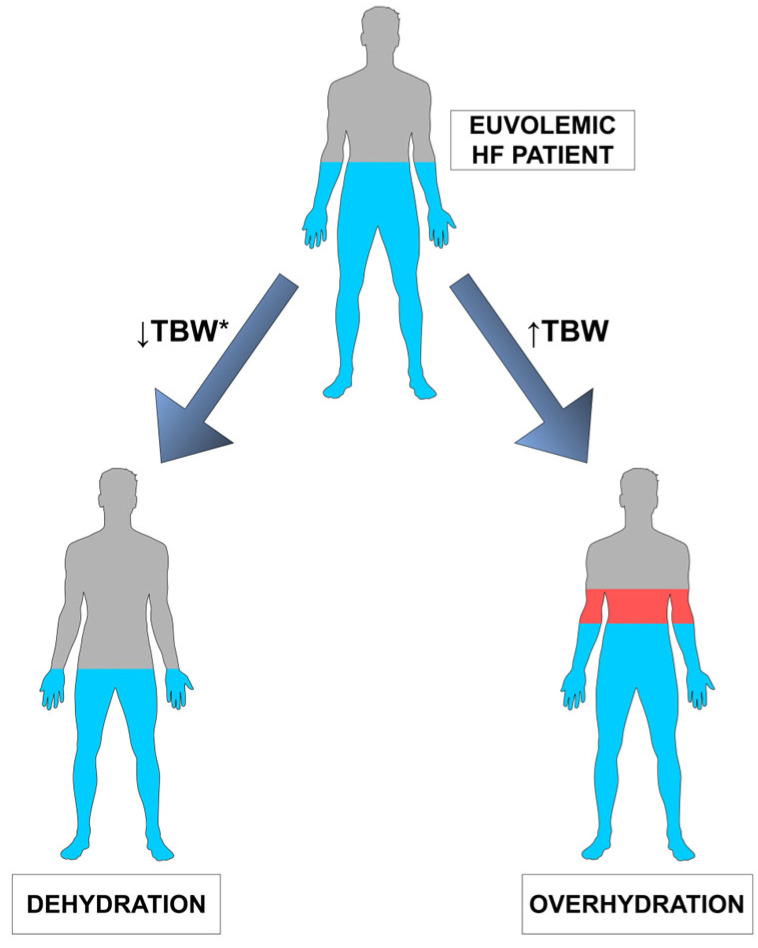
Simplified representation of the relationship between euvolemia, dehydration, and overhydration (congestion) in heart failure; TBW—total body water; *—Dehydration can be defined as a complex condition resulting in a reduction in total body water, however, the problem is more complex (see Section 4).

**Figure 2 biomedicines-11-02684-f002:**
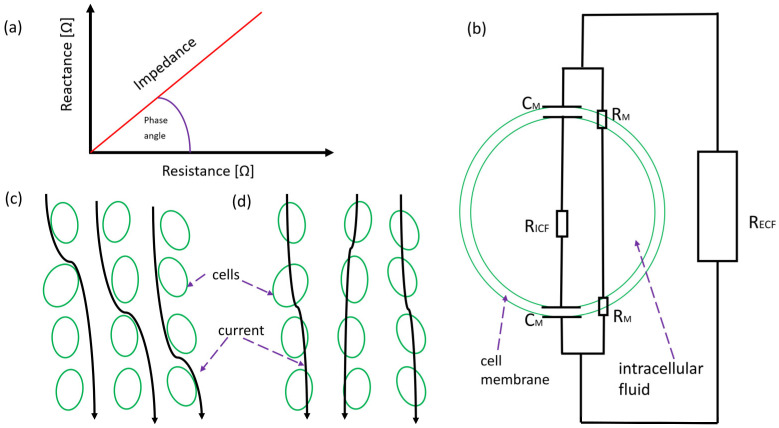
(**a**) Graphical definition of electrical impedance; (**b**) Equivalent circuit of human tissues according to Hayden model: CM—capacitance of cell membrane, RM—resistance of cell membrane, RICF—resistance of intracellular fluid, RECF—resistance of extracellular fluids; (**c**) Schematic of the flow of current through tissues for signal in range of kHz; (**d**) Schematic of the flow of current through tissues for signal in range of MHz, which according to simulations is more “direct” than in kHz frequencies.

**Figure 3 biomedicines-11-02684-f003:**
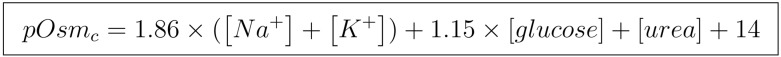
Khajuria and Krahn equation, recommended for calculating plasma osmolality. All variables have to be input in mmol/L. Abbreviations: *pOsm_c_*—calculated plasma osmolality; [*Na*^+^]—serum sodium concentration; [*K*^+^]—serum potassium concentration; [*glucose*]—serum glucose concentration; [*urea*]—serum urea concentration.

**Figure 4 biomedicines-11-02684-f004:**
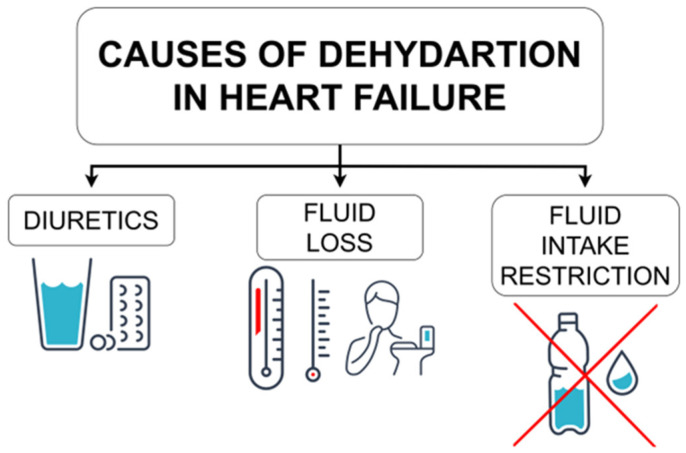
Possible causes of dehydration in heart failure.

**Figure 5 biomedicines-11-02684-f005:**
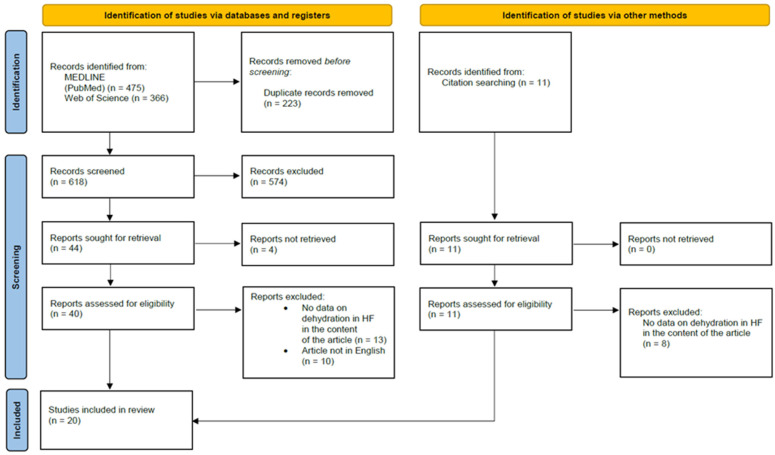
PRISMA flow diagram for the databases search.

**Figure 6 biomedicines-11-02684-f006:**
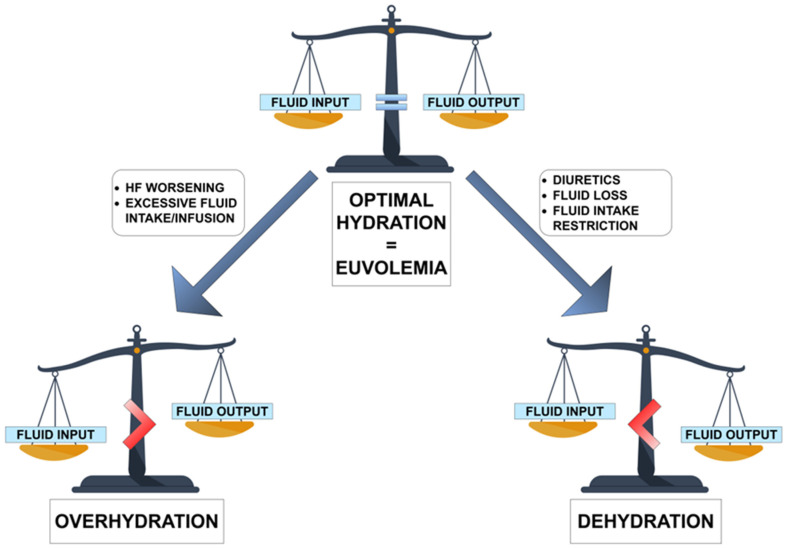
The graphical representation of relationships between euvolemia, overhydration, dehydration, and fluid input/output.

**Table 1 biomedicines-11-02684-t001:** Main clinical findings of dehydration and overhydration (congestion).

Clinical Finding	Overhydration (Congestion)	Dehydration
Symptoms	Dyspnea (also paroxysmal nocturnal dyspnea), orthopnea, bendopnea, peripheral edema (ankle swelling)	Confusion, fatigue
Signs	Jugular venous distention, the third heart sound, pitting edema	Tongue dryness and furrows, dry mucous membranes, low urine output, speech difficulty, sunken eyes, low blood pressure, increased pulse rate
Laboratory findings	Elevated circulating levels of natriuretic peptides	Elevated plasma osmolality (direct laboratory measurement or calculated); creatinine, urinary sodium, BUN (blood urea nitrogen)/urea
Non-invasive methods	Bioelectrical impedance analysis (BIA) and bioelectrical impedance vector analysis (BIVA); lung ultrasound (LUS); chest X-ray; chest computed tomography (CT)	Tracer dilution techniques for total body water (TBW) measurement (gold standard); bioelectrical impedance analysis (BIA), especially resistance at 50 kHz
Invasive methods	Cardiac catheterization—measurement of the right atrial pressure and pulmonary capillary wedge pressure (PCWP)	-

**Table 2 biomedicines-11-02684-t002:** A summary of four case reports on dehydration in heart failure identified in the databases during the search.

-	Case 1 [113]	Case 2 [114]	Case 3 [115]	Case 4 [116]
Date of publication	2022	2021	2013	2011
Title	Short-Term Treatment with Empagliflozin Resulted in Dehydration and Cardiac Arrest in an Elderly Patient with Specific Complications: A Case Report and Literature Review	Recurrent Takotsubo syndrome complicated with ischemic enteritis successfully treated by hydration: a case report	Paradoxical Heart Failure Precipitated by Profound Dehydration: Intraventricular Dynamic Obstruction and Significant Mitral Regurgitation in a Volume-Depleted Heart	Dehydration with High Natriuretic Peptide Levels! A Word of Caution
Authors; country	Supakul et al.; Japan	Shunsuke Todani and Mao Takahashi; Japan	Kim et al.; South Korea	Alzand et al.; The Netherlands
Age and sex of patient	68 years; man	80 years; woman	59 years; woman	81 years; woman
Clinical presentation	The patient was hospitalized for a cerebral infarction. During hospitalization, he was diagnosed with type 2 diabetes mellitus and HF (ejection fraction of 39% with global hypokinesia). Empagliflozin 10 mg/day was started to treat hyperglycemia and HF. Coronary angiography was performed on day 18 and showed 90% stenosis of the left anterior descending artery. A drug-eluting stent was deployed on day 31 to treat the lesion. During hospitalization, the patient experienced decreased appetite. On day 33, intravenous hydration was discontinued. On day 40, the patient had a cardiac arrest. The ECG showed asystole, and the patient did not respond to cardiopulmonary resuscitation. The cause of death was suspected to be related to dehydration due to low food and fluid intake associated with empagliflozin treatment, which may have led to acute kidney injury, hyperkalemia, and subsequent cardiac arrest.	The patient was admitted to the hospital with upper abdominal pain and bloody stools. She had a history of Takotsubo syndrome (TTS) complicated by ischemic enteritis 4 months earlier. Blood work showed that the patient’s brain natriuretic peptide (BNP) level had increased to 1578 pg/mL. Echocardiography showed wall motion abnormality centered on the left central ventricle with apical ballooning. A recurrence of TTS was diagnosed. The authors suspected that the abdominal pain and dehydration due to ischemic enteritis may have contributed to the development of TTS. A coronary angiography and an acetylcholine provocation test were conducted. No significant coronary artery stenosis was found, but the acetylcholine provocation test revealed significant multivessel coronary spasm in the left coronary artery, suggesting coronary vasospastic angina pectoris associated with TTS.	The patient presented with dyspnea (NYHA III/IV), chest radiography showed pulmonary congestion. Transthoracic echocardiography revealed systolic anterior motion (SAM) of the mitral valve (MV), resulting in dynamic left ventricular outflow tract (LVOT) obstruction and significant mitral regurgitation. The hemodynamic abnormalities were exacerbated by dehydration, which resulted in an acute decrease in cardiac output and hemodynamic compromise. Despite pulmonary edema, the authors decided to restore the patient’s systemic volume because the cause of pulmonary congestion was not due to absolute systemic volume overload, but to intracardiac volume maldistribution resulting from SAM of the MV in the volume depleted heart.	The patient presented to an outpatient clinic with complaints of palpitations. She was taking chlorthalidone 50 mg and atenolol 100 mg. Physical examination revealed sinus tachycardia of 120/min. Cardiac echocardiography showed systolic anterior motion of the mitral valve (MV) with a maximal gradient of 82 mmHg over the left ventricular outflow tract and a grade 3+ MV regurgitation (MR). Laboratory findings revealed a high NT-proBNP level of 1357 pmol/L, which was significantly elevated compared to the routine chemistry check-up 4 weeks prior to presentation. Despite the MR, there was no evidence of increased filling pressures. The patient was diagnosed with dehydration and admitted to the hospital, where diuretics were discontinued and she received an intravenous infusion of 1.5 L/day.
Cause of dehydration	Decreased appetite, inadequate fluid intake, discontinuation of IV fluids	Ischemic enteritis, hematochezia	The patient had not taken any food or water for a week due to longstanding anorexia and vomiting associated with her brain disease (medical history of old cerebrovascular accident and epilepsy).	Use of diuretics (chlorthalidone) and presumably inadequate fluid intake
Treatment	See above	Oxygen (2 L/min); 10,000 units/day of continuous intravenous heparin for 2 days; fluid replacement of 1500 mL/day to treat TTS	Fluid therapy with crystalloid and other nutritional support for five days.	Diuretics were ceased and the patient received 1.5 L/day intravenous infusion of 0.9% normal saline.
Follow-up	-	The BNP level and myocardial wall motion were normalized on the fourth day after admission. Subsequent follow-up showed no recurrence of takotsubo syndrome over 3 years.	After receiving fluid therapy, the patient’s condition stabilized and laboratory findings gradually returned to normal.	The NT-proBNP level normalized and the clinical as well as the echocardiographic parameters stabilized 3 days after treatment.
Author’s conclusions	“Although there are many advantages to using SGLT-2 inhibitors, it is important to note that there are individual risks that potentially lead to serious adverse effects, as shown in this case. Careful monitoring of elderly patients with neurological deficits who receive this medication is strongly recommended”.	“Our treatment of TTS in the acute phase centered on replenishment of fluid to increase coronary blood flow, improve heart load without exacerbating heart failure, and stabilize circulation dynamics. We recommend that clinicians consider treating TTS using hydration rather than diuretics in patients with dehydration”.	“The case provides helpful insights into intracardiac hemodynamics in a volume-depleted heart”.	“The present case report may have significant clinical implication, illustrating that elevated BNP or NT-proBNP levels should not robotically be assumed to be related to high filling pressures and reflexively resulting in intensifying diuretic therapy”.

## Data Availability

Not applicable.

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
