# Peer review of "The Importance of Optimal Hydration in Patients with Heart Failure—Not Always Too Much Fluid"

_biomedicines, 2023, doi:10.3390/biomedicines11102684_

Round 1

Reviewer 1 Report

The review targets an important issue in heart failure management - a balance between decongestion therapy and the risk of dehydration and could be interesting to potential readers.

However, the authors must address several critical issues and include more information before this review might be considered for publication.

  1. Edema and/or congestion rather than simple fluid overload are the major hallmarks of heart failure. 

Bozkurt, B.; Coats, A.J.S.; Tsutsui, H.; Abdelhamid, C.M.; Adamopoulos, S.; Albert, N.; Anker, S.D.; Atherton, J.; Bohm, M.; Butler, J.; et al. Universal Definition and Classification of Heart Failure: A Report of the Heart Failure Society of America, Heart Failure Association of the European Society of Cardiology, Japanese Heart Failure Society and Writing Committee of the Universal Definition of Heart Failure: Endorsed by the Canadian Heart Failure Society, Heart Failure Association of India, Cardiac Society of Australia and New Zealand, and Chinese Heart Failure Association. Eur. J. Heart Fail. 202123, 352–380. 

The terms "edema/congestion" and "fluid overload" are related but refer to different conditions. Fluid overload refers to the excess of total body water, while edema/congestion in heart failure is defined as fluid accumulation in the intravascular compartment and interstitial space. As such, the term "fluid overload' must be avoided. Please correct this terminology.

22. As dysregulation of the neurohumoral system, including the sympathetic nervous system, renin-angiotensin-aldosterone system (RAAS), and the natriuretic peptide system, crucially contribute to deregulation of sodium-fluid homeostasis leading to fluid accumulation and edema/congestion in heart failure, role, and mechanisms of these systems contribution to HF-related edema/congestion must be described in the review.

33.  Data support SGLT2 inhibitors as medication that may provide effective decongestion, suppressing cardiogenic edema of heart failure patients without side effects associated with diuretics, significantly reducing HF-related hospitalization/rehospitalization rate, and improving other heart failure outcomes.

Reviewer 2 Report

The authors wrote a review article entitled “the importance of optimal hydration in patients with heart failure- not always too much fluid” about the clinical implication of euvolemic status by avoiding hypervolemic and hypohydronic status in patients with heart failure.

1. The novelty of this review article would be the clinical implication of hypohydronic status in patients with heart failure. The overall text may be too much, and the paragraphs about hypervolemia may be shortened.

2. The focus of this review would be dehydration in patients with heart failure. Description of dehydration in general population may also be shortened to clarify their argument. 

na

Reviewer 3 Report

1.       I would recommend adding a table summarizing the main clinical findings of over-hydration and de-hydration.

2.       I would also recommend re-arranging the presentation in paragraph “3.1. Diagnostics”; it is rather confusing going from PCWP assessment to clinical findings and then to biompendance. You could just separate in sub-sections, such as clinical findings, imaging findings, laboratory markers, etc.

3.       I think that the paragraph “3.2 Treatment overview” should be substantially expanded; there is a vast amount of information now, especially for sacubitril/valsartan, SGLT-2 inhibitors and MRAs, in different HF settings.

4.       I would recommend omitting paragraph “5.2.2. Case reports” and all relevant information.

Minor revision is required.

Utilize the abbreviation "HF" consistently throughout the text.

Round 2

Reviewer 1 Report

Dear Authors,

Thank you for your point-to-point response to my comments. Still, several important statements related to congestion development and resolution in HF require clarification.

  1. Abstract:

Lines 15-16: “Although congestion is inextricably linked to heart failure, dehydration is also possible in HF patients.” – This statement is better removed from the Abstract, as congestion is not only linked to HF but rather the primary feature of HF. The statement on lines 16-17 delivers the same message.

Line 25: “Heart failure” needs to be abbreviated as “HF.”

2.       Major text:

As activation of the sympathetic nervous system (SNS) happened earlier than RAAS, subsection “3.1.2. Role of the sympathetic nervous system” should go before subsection “3.1.1. Role of renin-angiotensin-aldosterone system.”

The following milestone references must be incorporated to support the role of RAAS and NP hormones in HF:

- Schrier RW, Abraham WT. Hormones and hemodynamics in heart failure. N Engl J Med. 1999;341(8):577-85. 

- Weber KT. Aldosterone in congestive heart failure. N Engl J Med. (2001) 345(23):1689–97.

- Hartupee J, Mann DL. Neurohormonal activation in heart failure with reduced ejection fraction. Nat Rev Cardiol. (2017) 14(1):30–8. 

The subsection 3.1.1. Role of renin-angiotensin-aldosterone system:

- The crucial role of plasma renin activity in RAAS activation and modulation of edema/congestion in heart failure must be emphasized and adequately cited.

The subsection 3.1.3. Role of natriuretic peptide system:

Lines 165-167: This sub-section is superficially written. The major mechanisms related to the mentioned HF paradox include dysregulation of enzymatic activation/degradation pro-hormone/hormone mechanisms and receptor-recognition mechanisms:

  1. Impaired activation of pro-ANP/pro-BNP (pro-hormones forms) by cardiac expressed enzymes corin and furin.
  2. Elevated degradation of biologically active ANP and BNP by enzyme neprilysin.
  3. Impaired recognition of biologically active hormones ANP and BNP by their receptors on kidney cells.

- Ibebuogu UN, et al. Decompensated heart failure is associated with reduced corin levels and decreased cleavage of pro-atrial natriuretic peptide. Circ Heart Fail. (2011) 4(2):114–20.

- Costello-Boerrigter L, Lapp H, Boerrigter G, et al. Secretion of Prohormone of B-Type Natriuretic Peptide, proBNP1-108, Is Increased in Heart Failure. JACC: Heart Failure 2013;1(3): 207–212.

- McKie, P.M., Burnett, J.C. Rationale and Therapeutic Opportunities for Natriuretic Peptide System Augmentation in Heart Failure. Curr Heart Fail Rep 12, 7–14 (2015).

- Volpe M, Carnovali M, Mastromarino V. The natriuretic peptides system in the pathophysiology of heart failure: from molecular basis to treatment. Clin Sci. (2016) 130(2):57–77. 

- Zaidi SS, Ward RD, et al. Possible enzymatic downregulation of the natriuretic peptide system in patients with reduced systolic function and heart failure: a pilot study. Biomed Res Int. (2018) 2018:7279036. 

- Gladysheva I.P. et al. Falling corin and ANP activity levels accelerate the development of heart failure and cardiac fibrosis. Front. Cardiovasc. Med., 18 April 2023, 10; 2023.

Lines 335-336: the following statement is incorrect - “These drugs also reduce congestion by blocking the RAAS or inducing natriuresis.”

 It was reported that SGLTi prevents or reduces congestion through a comprehensive mechanism different from diuretics and RAAS system suppressants. Thus, SGLT2i not only stimulates diuresis/natriuresis but also reduces tissue edema. Please edit the referred statement and subsection 3.3.3. accordingly.

- Hernandez M, Sullivan RD, et al. Sodium-Glucose Cotransporter-2 Inhibitors Improve Heart Failure with Reduced Ejection Fraction Outcomes by Reducing Edema and Congestion. Diagnostics. 2022; 12(4):989.

- Sullivan RD, McCune ME, et al. Suppression of Cardiogenic Edema with Sodium–Glucose Cotransporter-2 Inhibitors in Heart Failure with Reduced Ejection Fraction: Mechanisms and Insights from Pre-Clinical Studies. Biomedicines2022; 10(8):2016. 

- Biegus J, Fudim M, Salah HM, et al. Sodium-glucose cotransporter-2 inhibitors in heart failure: Potential decongestive mechanisms and current clinical studies. Eur J Heart Fail. 2023.

The mechanism of Ang II modulation in homeostasis and HF must be clarified. Thus, Ang II regulates renin secretion through a homeostatic mechanism - the “short feedback loop.” Thus, renin synthesis and secretion are stimulated by reductions in Ang II concentration. When used chronically, Ang II converting enzyme inhibitors (ACEI) or Ang II receptor blockers (ARB) cause loss of negative feedback inhibition of renin production and stimulate renin production and secretion.

- Hackenthal E, Paul M, Ganten D, Taugner R: Morphology, physiology, and molecular biology of renin secretion. Physiol Rev 70: 1067–1116, 1990.

- Chen L, Kim SM, Eisner C, et al. Stimulation of renin secretion by angiotensin II blockade is Gsalpha-dependent. J Am Soc Nephrol. 2010 Jun;21(6):986-92. 

 Minor:

Abbreviation: The term “heart failure” needs to be consistently used in the abbreviated form of “HF” (for instance, on lines 53, 73, 323).
